# Effect of Microwave and Ultrasound-Assisted Extraction on the Phytochemical and In Vitro Biological Properties of Willow (*Salix alba)* Bark Aqueous and Ethanolic Extracts

**DOI:** 10.3390/plants12132533

**Published:** 2023-07-03

**Authors:** Ricardo S. Aleman, Jhunior Marcia, Carmen Duque-Soto, Jesús Lozano-Sánchez, Ismael Montero-Fernández, Juan A. Ruano, Roberta Targino Hoskin, Marvin Moncada

**Affiliations:** 1School of Nutrition and Food Sciences, Louisiana State University Agricultural Center, Baton Rouge, LA 70802, USA; 2Faculty of Technological Sciences, Universidad Nacional de Agricultura Road to Dulce Nombre de Culmí, Km 215, Barrio El Espino, Catacamas 16201, Honduras; 3Department of Food Science and Nutrition, University of Granada, Campus Universitario s/n, 18071 Granada, Spain; 4Department of Agricultural and Forestry Engineering, Escuela de Ingenierías Agrarias, Universidad de Extremadura, vda. Adolfo Su’arez s/n, 06007 Badajoz, Spain; 5C.I. Nutreo S.A.S., Iluma Alliance, Medellin, Colombia; juan.ruano@nutreo.co; 6Department of Food, Bioprocessing & Nutrition Sciences, Plants for Human Health Institute, North Carolina State University, North Carolina Research Campus, Kannapolis, NC 28081, USA

**Keywords:** phytochemistry, antioxidant, anti-inflammatory, medicinal plants, acetylcholinesterase inhibition

## Abstract

White willow (*Salix alba*) is a medicinal plant used in folk medicine. In this study, aqueous and ethanolic willow bark extracts were obtained via ultrasonic-assisted extraction (UAE) and microwave-assisted extraction (MAE), and analyzed regarding their phytochemical (total phenolics, phenolic acids, flavonoids, and tannins) content and in vitro biological properties (antibacterial and antifungal activity, acetylcholinesterase AChE inhibitory activity and anti-inflammatory effects). The highest phenolic, tannin, and flavonoid contents were found for willow bark extracts obtained via microwave-assisted extraction using ethanol as a solvent (SA-ME). The polyphenol load of all MAE and UAE extracts was higher when conventional solid–liquid extraction was applied (*ρ* < 0.05). The antioxidant capacities were stronger for microwave-assisted ethanolic extracts, with the lowest IC_50_ values of 12 μg/mL for DPPH^•^ and a value of 16 μg/mL for ABTS•+, whereas the conventional extraction had the highest IC_50_ values (22 μg/mL and 28 μg/mL, respectively). Willow bark extract showed antibacterial activity against Gram-positive bacteria *S. aureus* and *P. aeruginosa.* AChE inhibitory activity was dependent on the extraction method and solvent used, and the highest inhibition among samples was observed for SA-ME. Taken altogether, our findings suggest that willow (*Salix alba)* bark extract obtained via ethanolic microwave-assisted extraction is a phytochemical-rich resource with in vitro, anti-inflammatory, and AchE inhibitory properties and, therefore, potential multiple medicinal end-uses.

## 1. Introduction

The use of plant sources for medicinal purposes has been recorded since ancient times, their beneficial effects being attributed to their secondary metabolite composition. Thus, medicinal plants have proved to be incredible sources of bioactive molecules of great relevance and have been extensively studied as prime sources for the food, cosmetic, and pharmaceutical industries. Indeed, products of natural origin and related derivatives represent more than half of the Food and Drug Administration (FDA)-approved drugs [1]. In this sense, phenolic compounds have gained recent interest for their versatile health benefits, including their antioxidant, antimicrobial, and antiviral properties. The revalorization of plant residues as a source of these bioactive compounds is currently on the rise, as is the case of *Salicornia*, a halophyte plant that grows in Mediterranean areas, used not only as a vegetable, but also in traditional medicine for its beneficial effects against certain diseases and for its high level of bioactive compounds such as polyphenols [2].

Among medicinal plants with interesting bioactive content, the genus *Salix* (willow) presents great potential, comprising about 500 species that grow well in marginal wet land distributed worldwide. This large number of species makes this plant one of the most abundant sources of renewable raw materials for multiple end-uses [3,4,5,6]. Willow extracts and herbal tea derived from the plant bark are currently available from the species *S. purpurea, S. daphnoides*, and *S. fragilis* [7]. Willow bark (*Salix* spp., Salicaceae) has been recognized for its health-related attributes, namely, its antiproliferative, analgesic, antipyretic, and anti-inflammatory properties [8,9] due to its rich phytochemical composition, constituting mainly of antioxidant phenolic compounds [10,11]. In fact, phenolic acids such as salicylic acid are found in willow species, and its acetylated derivative (acetylsalicylic acid, Aspirin^®^) is one of the most successful examples of natural molecules used to develop profitable drugs in the history of phytomedicine [12,13,14]. Thus, the use of residues from these plants to obtain phenolic-rich extracts has proven to be quite promising for the exploitation of their bioactive potential.

Species like *S. alba* are reported to be of medical interest, but they are less studied and, as a result, information about them is scarce [10]. Extraction is one of the key steps for the production of herbal products, as it greatly influences the quality of the final plant extract and its final phytochemical content. Conventional techniques such as solid–liquid extraction using large amounts of harmful solvents continue to be widely used, but there is a solid interest in developing modern, faster, and more efficient and environmentally friendly strategies applied to the extraction of natural compounds. These so-called “classic” techniques are being progressively substituted with more sustainable extraction procedures such as ultrasound-assisted extraction (UAE) and microwave-assisted extraction (MAE) [15,16], due to profitability and bioactive recovery. In this regard, UAE is an extraction technique conducted at lower temperatures using reduced solvent volumes, which results in a final product with enhanced quality [17]. Likewise, MAE has several advantages such as reduced extraction time, enhanced extraction rate and yields, and can be developed using only water as the solvent [18]. To date, there is no research focused on the thorough study of the bioactive potential of willow (*Salix alba*) bark phenolic-rich aqueous and ethanolic extracts obtained through UAE and MAE, as environmentally friendly techniques for the revalorization of this residual matrix.

Therefore, in the present study, UAE and MAE were applied as extraction techniques of great potential for obtaining phytochemical-rich extracts from willow bark from *S. alba* species and characterizing the final product regarding its phytochemical (total phenolics, phenolic acids, flavonoids, and tannins) content and in vitro biological properties (antibacterial and antifungal activity, acetylcholinesterase inhibitory activity, and anti-inflammatory effects). This study aims to unveil the best extraction procedure and type of solvent to maximize the phytochemical content and in vitro biological activities of willow bark as a potential natural source of high-value medicinal compounds.

## 2. Results and Discussion

### 2.1. Phytochemical Content

In this study, UAE and MAE extraction techniques using water and water/ethanol as solvents were applied to obtain phenolic-rich extracts of willow (*S. alba*) bark, and compared with a conventional extraction, which was carried out as a control.

The total polyphenols content of aqueous and hydroalcoholic extracts of willow (*S. alba*) bark extracts ranged from 147.6 ± 0.2 to 169.8 ± 0.9 GAE mg/g DW (Table 1), and they were similar to a previous study of ethanolic extract of *Salix alba* extracted conventionally [19]. The obtained TPC was higher in all the considered extractions when compared to the conventional technique (*ρ* < 0.05), exhibiting the improved extraction potential of the applied technologies in this plant matrix. Ultrasonic-assisted and microwave-assisted extractions proved to be efficient methods for the extraction of bioactive compounds from plant sources [20,21,22,23].

Different results were observed when considering combinations of extraction methodologies and solvents, decreasing the TPC in the following order: SA-ME ˃ SA-UE ˃ SA-MA ˃ SA-UA. Thus, a clear influence of the type of solvent was observed for TPC, since a better extraction performance was observed in hydroalcoholic extraction than aqueous. When using water as extraction solvent, no significant differences were observed between both extraction methodologies applied. However, the use of an aqueous ethanolic solvent leads to differences in extraction performance between UAE and MAE, with a higher phytochemical content being observed in the latter. These results are supported by the literature, where solvents used in ultrasonic-assisted and microwave-assisted extractions proved to play an important role on the performance of the extraction process [20]. Higher extraction yields of bioactive compounds from red oak bark have been shown when both UAE and MAE were conducted with an ethanolic solvent solution [19]. This has also been observed when sonicated extracts of willow bark presented a higher extraction of phytochemicals when using ethanol as a solvent compared with the use of water.

In addition, total tannin and flavonoid content was also evaluated. When the total tannin, phenolic acids, and flavonoid content of samples were compared, SA-ME showed again the highest results among samples (*ρ* < 0.05; Figure 1). Conventional extraction samples had the lowest results, comparable to SA-UA group, but SA-MA and SA-UE presented the best extraction performance (*ρ* > 0.05). In this case, influence of the combination of the selected solvent can also be observed, leading the use of aqueous ethanol to an increased extraction of all the considered phytochemicals, especially when combined with microwave-assisted extraction. This is consistent with previous literature, where the extraction yield of phenolic compounds proved to be higher in MAE than in UAE [20].

The present results exhibit the extraction improvement of hydroalcoholic microwave-assisted extraction compared to ultrasonic-assisted extraction. MAE utilizes microwaves to heat the sample solvent, rapidly partitioning the polyphenols from the ethanol. The use of microwaves has proved an enhancement from conventional extraction as microwaves are shown to interact with polar molecules in the extraction media [24]. Although multiple factors can impact the extractability of phenolic compounds [25], higher extraction has been proposed to be explained through the evaporation of cell moisture and its consequent pressure on the cell wall, that favors its rupture and release of bioactive compounds [26].

The presented MAE results are consistent with the previous literature for other phenolic sources. MAE extracts have proved a better yield of bioactive compounds from *Aloe vera* and *Hibiscus sabdariffa* when compared with the use of conventional extraction [27,28]. Furthermore, MAE has been extensively applied to recover bioactive chemicals from several plant matrices, leading to excellent phenolic compound recoveries [29].

Thus, the value of willow bark as a raw source of phenolic compounds can be enhanced through a rational extraction strategy, as demonstrated by our results, which support the advantages of MAE and UAE as advanced extraction techniques when compared to conventional extraction methods [8].

### 2.2. Antioxidant Activity

Many biological activities observed for medicinal plants are directly linked to the content of antioxidant polyphenols [30]. The DPPH^•^ radical scavenging activity and ABTS•+ examination of the willow bark aqueous and ethanolic extracts obtained by UAE and MAE are shown in Table 1. Once again, the radical scavenging activity of UAE or MAE samples demonstrated lower IC_50_ values than conventional extraction samples indicating the superior stronger antioxidant activity in these extracts (Table 1). Moreover, SA-ME treatments had the lowest IC_50_ values for both DPPH^•^ and ABTS•+ scavenging capacity (above 12 μg/mL and 16 μg/mL, respectively), indicating an outstanding antioxidant potential, whereas the conventional extraction had the highest IC_50_ values (22 μg/mL and 28 μg/mL, respectively). Our results show an expected relationship between TPC and antioxidant activity, which proves that phenolic compounds play an important role on increasing the in vitro antioxidant activity of willow bark extracts, similarly to what was shown in a previous report [31] on red oak barks.

The *Salix* family contains notable amounts of endogenous salicylate compounds and excellent antioxidant capacity. A *S. alba* bark extract has been reported to have the highest radical scavenging activity against DPPH^•^ among the seven analyzed bark extracts [20]. Additionally, other species from the studied family have also presented significant activities. The leaf extract of *S. triandra* exhibited the most potent DPPH^•^ quenching capacity compared to other leaf extracts from *S. amplexicaulis, S. babylonica, S. eleagnos, S. fraglis, S. purpurea,* and *S. alba* [20]. Similar results have been observed for *Salix eleagnos* bark hydroethanolic extracts, where MAE extracts showed a lower IC_50_ when compared to UAE, supporting the observed results in this study [20] (1.98 vs. 2.23; IC_50_ μg/mL). In leaf extracts of *Salix mucronata Thunb.,* MeOH (85%) extract showed high antioxidant activities, with higher IC_50_ values than observed in the present study, emphasizing the potential of the selected matrix and solvent (DPPH^•^ IC_50_= 98.76 ± 0.46 (µg/mL), ABTS•+= 45.83 ± 0.32 mg Trolox^®^ eq. /100 mg extract and TAC= 199.18 ± 2.19 mg equivalent of ascorbic acid/g ext.) [32]. Our results for UAE extractions are also promising when compared with the reported antioxidant activities for *Salix alba* leaves in the study by Piatczak et al. (2020) [11], where higher IC_50_ concentrations were observed in leaves (DPPH^•^ SC_50_= 28.23 (µg/mL), ABTS•+ = 65.43 (µg/mL)) and bark (DPPH^•^SC_50_= 13.51 (µg/mL), ABTS•+ = 21.50 (µg/mL)) for UAE. Overall, our results indicate a high antioxidant capacity of all obtained *Salix alba* bark extracts, indicating an improvement in this parameter when MAE ethanolic extracts were selected.

### 2.3. Antibacterial and Antifungal Activities

Antimicrobial activity is one of the many mechanisms by which phenolic-rich plants can exert their health-related activities [33]. A well diffusion assay was used to determine the inhibition zones of the *S. alba* bark extracts (aqueous/hydroalcoholic) obtained by UAE and MAE. Results (Table 2) show antibacterial activity against two Gram-positive bacteria (*S. aureus* and *P. aeruginosa*), but no antimicrobial effect of willow bark extracts was detected against Gram-negative bacteria (*E. coli* and *K. pneumoniae*) and yeasts (*A. terreus* and *R. stolonifera*) tested in this study. This observed effect could be related to the differential susceptibility of Gram-positive and Gram-negative bacteria to the antimicrobial effect of bioactive phytochemicals. Gram-negative bacteria can be more resistant to phenolics than Gram-positive bacteria due to their rigid cell wall structure and strong electronegativity of its outer membrane, which stablishes weaker interactions with phenolic molecules and jeopardize the effect of antimicrobial compounds, including antibiotics [34,35]. Our results agree with those of Sulaiman et al. (2013) [19], who also reported that *E. coli* and *K. pneumoniae* were not affected by *S. alba* extracts [36], where phytochemicals extracted from in eight *Salix* species showed no antifungal activity. An antibacterial effect on *S. aureus* of tannin-rich *S. alba* bark extracts has also been reported in previous literature [37]. Tannins have significant antifungal [38] and antibacterial properties that facilitate the extract to penetrate and exert its antimicrobial effects [39] *S. aureus* and *P. aeruginosa* are sensitive to polyphenols such as galangin, apigenin, naringenin, (−)-epigallocatechin-3-gallate, and phenolic acids [40]. The higher tannin content of willow bark ethanolic SA-ME extracts (Figure 1) might play a role in the observed antibacterial results (Table 2). Similarly, other studies have reported that microwave-assisted extraction improved antibacterial activity in waste peanut shells, olive tree leaves, and blackcurrant powder [41,42,43]. Overall, the present results indicate that SA-ME extracts can improve the antimicrobial activity against *S. aureus* and *P. aeruginosa*.

### 2.4. AChE Inhibition Activity of Willow (Salix alba) Bark Extracts

Acetylcholinesterase (AChE) is one of the types of cholinesterase, an enzyme needed for the central nervous system to properly function [44,45]. The main physiological function of AChE is to hydrolyze acetylcholine in the synapse and neuromuscular junction, which results in important neurological damage [46]. Cholinesterase inhibitors hinder the AChE activity and, therefore, maintain acetylcholine levels by mitigating its breakdown rate [20].

In the present study, the inhibitory potential of *Salix alba* (L.) bark extracts (aqueous/hydroalcoholic) obtained by UAE and MAE against AChE activity was investigated, using galantamine as the positive control. Both UAE and MAE *S. alba* bark extracts showed higher AchE inhibition (*ρ* < 0.05) compared to samples processed via conventional solid–liquid extraction. IC_50_ values varied depending on the extraction method and solvent used and the highest inhibition among samples was observed for SA-ME (Figure 2). In fact, willow bark extracts had lower IC_50_ values when compared to other medical plants, such as *Scolopia crenata* bark (IC_50_ = 9.20 μg/mL) [47] and *Viburnum lantana* extract (IC_50_ value = 50 μg/mL) [48]. Among the *Salix* family, the presented results resulted in a higher anticholinesterase activity than *S. amplexicaulis* bark and leave extracts and were comparable to a previous study on *S. alba* by Zaiter et al. (2016) [49]. Zaiter et al. (2016) [49] noted the inhibitory activity of *S. alba* bark extracts in AChE. The AChE IC_50_ values are associated with bioactive compounds such as chlorogenic acid, catechin, gallocatechin, salicin, and procyanidins B1, B2, and C1 [20]. The higher recovery of these bioactive compounds in *S. alba* bark extracts could improve the anticholinesterase activity. Taken altogether, the in vitro results of AChE inhibitory activity indicate the potential of SA-ME extracts as a possible natural resource for the mitigation improvement of the neurological effects of Alzheimer’s disease.

### 2.5. Anti-Inflammatory Activities of Willow (Salix alba) Bark Extracts

The anti-inflammatory activity of *Salix alba* (L.) bark extracts was estimated via the Caco-2 cells monolayer model. The cells were treated during 72 h with control vehicle (growth media, C) and inflammatory stimulus (I) consisting of 25 ng/mL IL-1β, 50 ng/mL TNF-α, 50 ng/mL IFN-γ, and 1 μg/mL LPS after pre-treatment with *Salix alba* extract samples (10 µg/mL) for 30 min. Cells were treated with SA-C, SA-ME, SA-MA, SA-UE, or SA-UA extracts. The trans-epithelial electrical resistance (TEER) and the passage of Lucifer Yellow (LY) and fluorescein isothiocyanate dextran (FD) were measured to evaluate Caco-2 cells’ integrity, as illustrated in Figure 3 and Figure 4. When the cells were treated with inflammatory stimulus, lower TEER values (<200 Ω * cm^2^) were observed when compared with untreated cells (control), and higher active transport rates for LY and FD were obtained that indicate structural disruption of tight junctions structural disruption, followed by reduced barrier function, resulting in inflammation. In contrast, all UAE and MAE *S. alba* bark extracts displayed higher TEER values, and low active transport rates (FD and LY) compared with cells only treated with the inflammatory stimulus after 72 h of treatment. The TEER results and LY and FD flux indicate that *S. alba* bark extract enables proper epithelial barrier function and communication between neighboring cells, which results in anti-inflammatory efficacy. The results indicated that when Caco-2 cells were treated with SA-C, SA-ME, SA-MA, SA-UE, and SA-UA extracts for 48 h (Figure 3b), none of the *Salix alba* extracts obtained via UAE or MAE negatively affected the monolayer TEER measurements. SA-ME and SA-UE samples had significantly (*ρ* < 0.05) lower FD flux, while only the former had significantly (*ρ* < 0.05) lower LY flux and higher TEER when compared with SA-C. Thus, administration of the obtained extracts results in an improvement in the inflammatory state of Caco-2 cells, with the SA-ME extract being the most promising. Other species of *Salix*, such as *S. pentandra* extracts, were reported to suppress cytokines (L-6, IL-1β, IL-10), and prostaglandin E2 [50]. The anti-inflammatory properties of white willow bark extracts are mainly due to the downregulation of the pro-inflammatory effects of NF-κB, COX-2, and TNF-α [51]. In this study, our hypothesis is that the anti-inflammatory activity of the SA-ME was primarily due to soluble compounds in ethanol. The phenolic compounds, primarily simple phenolics and tannins, are highly dissolved in ethanol, and their content was mostly improved by MAE, as confirmed in this study (Table 1, Figure 1). Indeed, tannins and simple phenolic metabolites have been shown to have anti-inflammatory activity in Caco-2 cells in previous studies [52,53]. In macrophage cells (RAW 264.7), MAE improved the ability to contain lipopolysaccharide-induced reactive oxygen species and produce nitric oxide in Algerian olive stone extracts [54]. Overall, SA-ME treatment had a significantly lower (*ρ* < 0.05) permeability when compared with all other treatments (SA-C, SA-MA, SA-UE, and SA-UA) regarding the FD flux (Figure 4b).

## 3. Materials and Methods

### 3.1. Materials

White willow bark (*Salix alba L*.) was purchased from (Herbal Tea Health Embassy), (Gloucestershire, UK). All reagents were analytical grade and supplied as follows: Folin–Ciocâlteu reagent (Sigma-Aldrich, St. Louis, MO, USA), Na_2_CO_3_ (Sigma-Aldrich, St. Louis, MO, USA), gallic acid (Sigma-Aldrich, St. Louis, MO, USA), pyrogallol (Sigma-Aldrich, St. Louis, MO, USA), phosphomolybdotungstic reagent (Sigma-Aldrich, St. Louis, MO, USA), diethylene glycol reagent diethylene (Sigma-Aldrich, St. Louis, MO, USA), NaOH (Fisher Chemical, Fair Lawn, NJ, USA), quercetin (Sigma-Aldrich, St. Louis, MO, USA), Sodium molybdate (Fisher Chemical Fair Lawn, NJ, USA), rosmarinic acid (Sigma-Aldrich, St. Louis, MO, USA), DPPH^•^ reagent (Sigma-Aldrich, St. Louis, MO, USA), Trolox (Sigma-Aldrich, St. Louis, MO, USA), nutrient agar (Difco, Detroit, MI, USA), potato dextrose broth (PDB, Difco, Detroit, MI, USA), Tris-HCl buffer (Sigma-Aldrich, St. Louis, MO, USA), Ellman’s Reagent (Sigma-Aldrich, St. Louis, MO, USA), enzyme AChE (Sigma-Aldrich, St. Louis, MO, USA),. Bovine serum albumin (BSA) (Sigma-Aldrich, St. Louis, MO, USA), galantamine (Sigma-Aldrich, St. Louis, MO, USA), acetylthiocholine iodide (Sigma-Aldrich, St. Louis, MO, USA), Dulbecco’s Modified Eagle’s medium (DMEM; Invitrogen, Waltham, MA), 10% FCS medium (Invitrogen, Waltham, MA), 1% non-essential amino acid (NEAA) medium (Invitrogen, Waltham, MA), penicillin (Fisher Chemical, Fair Lawn, NJ, USA), streptomycin (Fisher Chemical, Fair Lawn, NJ, USA), Trypsin-EDTA (Fisher Chemical, Fair Lawn, NJ, USA), IL-1 (Fisher Chemical, Fair Lawn, NJ, USA), IFN- (Fisher Chemical, Fair Lawn, NJ, USA), TNF- (Fisher Chemical, Fair Lawn, NJ, USA), and LPS (Fisher Chemical, Fair Lawn, NJ, USA).

### 3.2. Phytochemical Extraction

Initially, willow bark powder (WBP) was prepared as a source material for all the considered extractions. For this, white willow bark was dehydrated using a convection oven (50 °C for 30 min) (Digitronic TFT- Selecta, J.P. SELECTA, Barcelona, Spain), grinding the resulting product in a Retsch SM 100 knife mill (Retsch GmbH, Germany) (501–700 mm).

Figure 5 shows a schematic diagram of the experimental plan. Five willow bark extracts were obtained from applying different extraction techniques and conditions on the previously described WBP: using conventional solid–liquid extraction (SA-C), using ultrasound-assisted extraction with either water (SA-UA) or ethanol/water (50/50, *v*/*v*) (SA-UE) as extraction solvents, and by microwave-assisted extraction with water (SA-MA) or ethanol/water (70/50, *v*/*v*) (SA-ME) as the selected extraction solvents. All experiments were conducted in triplicate.

The conventional solid–liquid extraction samples (SA-C) were prepared by mixing 1 g WBP in 55 mL of water. The solution was continuously stirred using a magnetic stirring plate, heated at 55 °C for 37 min. Both ultrasound-assisted extraction (UAE) and microwave-assisted extraction (MAE) were performed following adapted protocols based on [31] since red oak and willow barks are similar matrices [41]. UAE was performed at 40 kHz ultrasonic power using an Elma Transsonics ultrasonic bath (Elma Schmidbauer GmbH, Singen, Germany). Initially, 2.5 g WBP was added to a volumetric flask with 100 mL water (SA-UA) or ethanol/water (50:50 *v*/*v*, SA-UE). The obtained solution was ultrasonicated for 15 min and heated at 70 °C [31]. using a water bath. For MAE, 10 g of WBP was added into the microwave extractor vessel with 200 mL of water (SA-MA) or ethanol/water (70:30 *v*/*v*, SA-ME). The extractions were performed in an Ethos X Advanced Microwave Extractor (Milestone, Sorisole, Italy). The aqueous extracts were obtained after 30 min at 850 W, while the hydroalcoholic extracts were obtained after 18 min at 650 W [31].

The conventional, UAE, and MAE extracts were filtered in a double layer of cheesecloth, and the filtrate was collected for further analysis. In addition, the filtrates of all extracts were frozen (−80 °C) and lyophilized (LIOTOP model L 101) at −75 ± 1 °C and 0.2 ± 0.1 Pa for 48 h. The lyophilized product was then ground in a commercial mill (LABOR model SP31). The filtrate powder (lyophilized extract) was vacuum-packed in plastic bags.

### 3.3. Phytochemical Characterization

#### 3.3.1. Total Phenolic Content (TPC)

Total phenolic content was determined via the previously reported Folin–Ciocalteu method [42] with few modifications. Initially, 550 µL of Folin–Ciocalteu reagent was added to 550 µL of diluted extract (15 mg/mL) followed by 3500 µL of 5% Na_2_CO_3_ solution. The tubes were thoroughly mixed and left in the dark at room temperature for 1 h. Using a Specord 200 Plus, the absorbance of samples was measured at 750 nm (Analytik Jena AG, Jena, Germany). The results were reported as mg gallic acid equivalent (GAE)/g sample DW.

#### 3.3.2. Phenolic Acids

Using the methodology of Bojic et al. (2013) [43] with slight modifications, total phenolic acids were determined by measuring the absorbance at 505 nm (Analytik Jena AG, Jena, Germany). Sodium molybdate (Fisher Chemical, Fair Lawn, NJ, USA) solution was used as the reference. A standard curve was created using a rosmarinic acid standard solution, and the results were expressed as rosmarinic acid equivalent (RAE)/g sample DW.

#### 3.3.3. Total Flavonoids Content (TFC)

A previously described colorimetric method with slight modifications was applied for the determination of the total flavonoid content [44]. Briefly, non-freeze-dried filtrate extracts (100 μL) were mixed with diethylene glycol (500 μL) and NaOH (50 μL, 1 N) and incubated at 37 °C for 60 min. Absorbance at 420 nm was measured using an ELISA reader (Synergy 2; BioTek Instruments). Using quercetin as the standard, the total flavonoid was expressed as mg of quercetin equivalent (QE)/g DW.

#### 3.3.4. Tannin Content

The tannin content was evaluated spectrophotometrically according to the European Pharmacopoeia 10.8 (Determination of tannins in hide powder). For this, the absorbance of samples was measured at 760 nm using a Specord 200 Plus (Analytik Jena AG, Jena, Germany). Results were represented as mg pyrogallol /g sample DW.

### 3.4. In Vitro Antioxidant Activity—Free Radical Scavenging Assays

#### 3.4.1. 2,2-Diphenyl-1-picrylhydrazyl (DPPH^•^) Assay

The test consisted of initially resuspending the lyophilized willow bark extracts obtained by conventional extraction, UAE and MAE (1 mg/mL) in the used extraction solvent. After homogenization, 200 μL of DPPH^•^ 0.1 mM was added to each well. After 30 min, the absorbance of each mixture was measured at 517 nm using a microplate reader (Epoch, BioTek, Winooski, VT, USA) and the antioxidant capacity was estimated according to Equation (1). [45] Trolox was used as a reference.
Inhibition (%) = 1 − (A_c_/A_s_) × 100(1)
where A_c_ is the absorbance of blank solution (solution without samples) and A_s_ is the absorbance of sample. Using a dose–response curve, the half maximum inhibitory concentration (IC_50_) was computed and represented as μg /mL. The IC_50_ was calculated using the normalized logarithmic curve for the determined percentages of inhibition, and the dependence between log C and % I was plotted using Prism 8. (GraphPad, San Diego, CA, USA).

#### 3.4.2. ABTS^•+^ (2,2′-Azino-bis (3-ethylbenzothiazoline-6-sulfonic acid) Assay

It was conducted according to a modified method by Laczkó-Zold et al. (2018) [46] using 96-well microplates. Initially, 55 μL of extract (1–0.550 mg/mL) or control (water) and 250 μL of ABTS^•+^ solution was transferred to each well. After allowing the mixture to react for 7 min, the absorbance at 734 nm was measured using a microplate reader (Epoch, BioTek, Winooski, VT, USA). The inhibition percentage and IC_50_ were calculated according to Equation (1).

### 3.5. Antimicrobial and Antifungal Activities

The antimicrobial activity of willow bark (*S. alba)* extracts obtained through conventional extraction, UAe and MAE was determined via the agar-well diffusion method. Bacterial strains (*Staphylococcus aureus* (methicillin-resistant) (ATCC 43300), *Escherichia coli*, (ATCC 25922) *Pseudomonas aeruginosa* (ATCC 9027), *Klebsiella pneumoniae* (ATCC 1388), *Shigella sonnei* (ATCC 25931)) and two yeasts (*Aspergillus terreus* (ATCC 20542) and *Rhizopus stolonifer* (ATCC 13429)) were used in present study. The antimicrobial activity was performed using nutrient agar for bacteria and potato dextrose agar medium for antifungal activity. The cell culture suspension was adjusted by comparing against 0.4–0.5 McFarland scale standard. Suspensions (100 mL) of each target strain were spread on the plates. For the investigation of the antimicrobial and antifungal activity, 100 µL of diluted willow bark extract (100 mg WBP/mL) was added into agar plate wells. The diameter of the inhibition zone (mm) was measured using the Aura Image software (Aura Image system, Tempe, AR). The incubation (bacteria at 37 °C, fungi at 28 °C) was conducted for 48 h. All samples were tested in triplicate.

### 3.6. Acetylcholinesterase (AChE) Inhibitory Activity

The inhibition of AChE was determined using a modified version of Ellman’s assay [47,48]. The reaction was initiated by combining diluted (serial dilutions between 4 and 0.25 mg/mL) willow bark extracts (25 μL) with Tris-HCl buffer (50 μL; 50 mM, pH 8.0), DTNB solution in Tris-HCl buffer (125 μL; 0.9 mM), and enzyme solution (25 μL; 0.078 U/mL), then incubating for 15 min at room temperature in the dark. Additional ATCI solution (25 μL; 4.5 mM in Tris-HCl buffer) was added, followed by a 10 min re-incubation of the samples. Samples absorbance was then measured at 405 nm wavelength using galantamine as a positive control. The percentages of inhibition (% I) were computed as Equation (2). Galantamine was used as the positive control.
Inhibition (%) = 1 − (A_r_/A_s_) × 100 (2)
where A_r_ represents the absorbance of the blank solution (solution without samples) and A_s_ represents the absorbance of the sample. Results were expressed as IC_50_ (the sample concentration that inhibited 50% of the enzyme, in μg /mL).

### 3.7. Anti-Inflammatory Activity

#### 3.7.1. Caco-2 Cells Maintenance

Human colon carcinoma cell lines Caco-2 (ACC169) were obtained from the European Collection of Authenticated Cell Cultures (Porton Down, UK). Separate flasks of cells were grown in Dulbecco’s Modified Eagle’s medium (DMEM, Invitrogen, Waltham, MA) supplemented with 10% FCS, 1% non-essential amino acid (NEAA), 100 U/mL penicillin, and 100 g/mL streptomycin at 37 °C in a humidified incubator containing 5% CO_2_/95% air. The culture medium was replaced every 2–3 days. Cells were periodically observed under a microscope and following differentiation until 80–90% confluence. The cells were kept at 37 °C in a humidified incubator containing 5% CO_2_/95% air [50].

#### 3.7.2. Caco-2 Cells Treatments

As described previously [49,51], monocultures of Caco-2 cells were harvested with Trypsin-EDTA and seeded on the apical chamber of 12-well ThinCert^®^ inserts (0.4 m PET pore membrane, Greiner Bio-One, Frickenhausen, Germany) to reach a final density of 3 × 105 cells/cm^2^ The cells were co-cultured for 19–21 days in a humidified incubator with a 5% CO_2_/95% air environment, with the apical and basolateral media changed every 2–3 days. After pre-treatment with Salix alba extract samples (10 µg/mL) (SA-C, SA-ME, SA-UE, SA-MA, and SA-UA) [28] for 30 min, cells were subjected to inflammatory co-stimulators including IL-1 (25 ng/mL), IFN-γ (50 ng/mL), TNF-α (50 ng/mL), and LPS (1 g/mL) for 72 h in a humidified incubator containing 5% CO_2_/95% air at 37 °C. The untreated cells (just treated with growth media) were considered as the negative control.

#### 3.7.3. Apparent Permeability Coefficient (Papp)

On days 19–21 of culture, the monolayer integrity of the Caco-2 co-culture was evaluated using transepithelial electrical resistance (TEER) measurements with an epithelial volt-ohmmeter equipped with “chopstick” electrodes (Millicell^®^ ERS, Millipore, Bedford, MA, USA) (Chelakkot et al., 2018). Co-culture inserts with TEER values above 200 cm^2^ were utilized. The literature-based flow of fluorescein isothiocyanate (FITC)-dextran 4000 (FD) and lucifer yellow (LY) through cell monolayers was used to assess paracellular permeability [52,53]. At 37 °C, FD (1 mg/mL) and LY (0.5 mg/mL) were combined with HBSS/HEPES and 0.2 mL was added to the apical compartment while 1 mL of HBSS was added to the basolateral compartment. The plates were incubated at 37 °C and 150 rpm while covered. Every 4 h for a total of 16 h, 300 μL of the basolateral chamber was transferred into a black, clear-bottomed 96-well plate (Corning Costar, New York, NY, USA). The basolateral compartment was then rinsed and 1 cc of fresh HBSS at 37 °C was added. Using a fluorescence spectrophotometer PLX800 (BioTek, Winooski, VT, USA) with excitation/emission wavelengths of 485/530 nm (FD) and 428/540 nm (EF), the fluorescence intensity was measured (LY). The apparent permeability coefficient (Papp) was calculated using the following formula: (Equation (3))
(3)$$P_{\mathrm{app}}\;(\mathrm{cm}/s)=\frac{\mathrm{dQ}}{\mathrm{dt}}\times \frac{1}{A\times C}$$
where dQ is the concentration of fluorescent markers on the basolateral side (mol/mL), dt is the function of time per second (1/s), A is the membrane’s surface area (cm^2^), and C is the initial concentration on the apical side (mol/mL). Results were expressed as cm/s.

### 3.8. Statistical Analysis

The statistical analysis was conducted using version 25 of IBM SPSS Statistics (IBM Corporation, New York, NY, USA). The data represent the means ± standard deviation (SD) of at least three independent experiments. The phenolic profile, antioxidant capacity, antibacterial potential, and enzymatic inhibition were analyzed using one-way ANOVA at *p* value < 0.05. For TEER and FD and LY Flux measurements, the generalized linear model was applied to determine the significance (*p* value < 0.05) between treatment effect, time effect, and interaction effect (treatment*time). Post hoc analysis was performed using the Dunnett’s multiple comparison tests.

## 4. Conclusions

This study showed that microwave-assisted and ultrasound-assisted extractions enhance the total phenolic content, antioxidant activity, antibacterial, enzymatic, and anti-inflammatory properties of *Salix alba* aqueous and ethanolic bark extracts, evidencing better results than previously reported for other species. The findings indicate that *Salix alba* aqueous ethanolic extract treated with microwave-assisted extraction might be a desirable strategy to obtain phytochemical-rich extracts with improved anti-inflammatory properties, pathogenic antibacterial potential, antioxidant capacity, and phenolic profile, and acetylcholinesterase inhibition activity. For future studies, the application of combined microwave and ultrasonic-assisted extractions on willow bark aqueous and ethanolic extracts should be considered to evaluate potential positive synergistic effects that can possibly maximize the phenolic content and antioxidant capacity, antibacterial potential, enzymatic inhibition, and anti-inflammatory properties.

## Figures and Tables

**Figure 1 plants-12-02533-f001:**
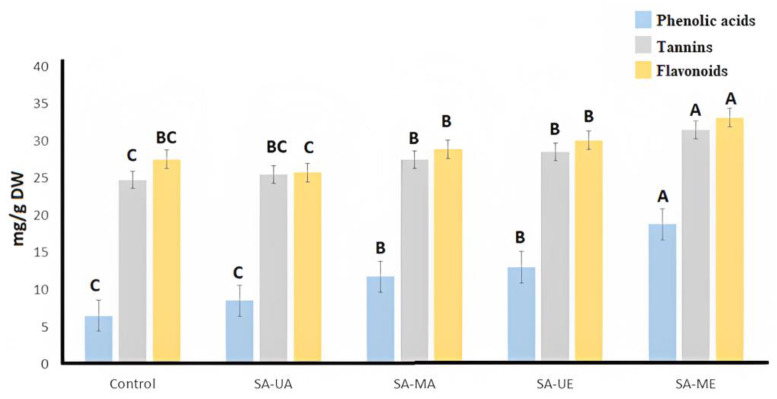
Phenolic acids, total tannin and total flavonoid contents (TFC) of willow (*Salix alba)* bark extracts. Results expressed as rosmarinic acid equivalent (RAE) /g sample DW (phenolic acids), mg pyrogallol /g sample DW (total tannin) and mg of quercetin equivalent (QE)/g DW (total flavonoids). Legend: Control: willow bark extract obtained via conventional solid–liquid extraction; SA-UA: willow bark extract obtained via ultrasound-assisted extraction with water; SA-MA: willow extract obtained via microwave-assisted extraction with water; SA-UE: willow bark extract obtained via ultrasound-assisted extraction with ethanol; and SA-ME: willow bark extract obtained via microwave-assisted extraction with ethanol. Bars with different letters differ significantly (*ρ* < 0.05).

**Figure 2 plants-12-02533-f002:**
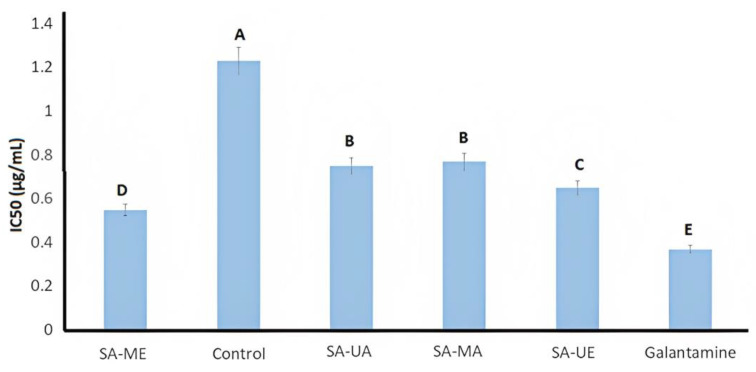
Anti-acetylcholinesterase (AchE) assay (results expressed as IC_50_) of willow (*Salix alba)* bark extracts and galantamine as positive control. Legend: Control: willow bark extract obtained via conventional solid–liquid extraction; SA-UA: willow bark extract obtained via ultrasound-assisted extraction with water; SA-MA: willow extract obtained via microwave-assisted extraction with water; SA-UE: willow bark extract obtained via ultrasound-assisted extraction with ethanol; and SA-ME: willow bark extract obtained via microwave-assisted extraction with ethanol. Bars with different letters differ significantly (*ρ* < 0.05).

**Figure 3 plants-12-02533-f003:**
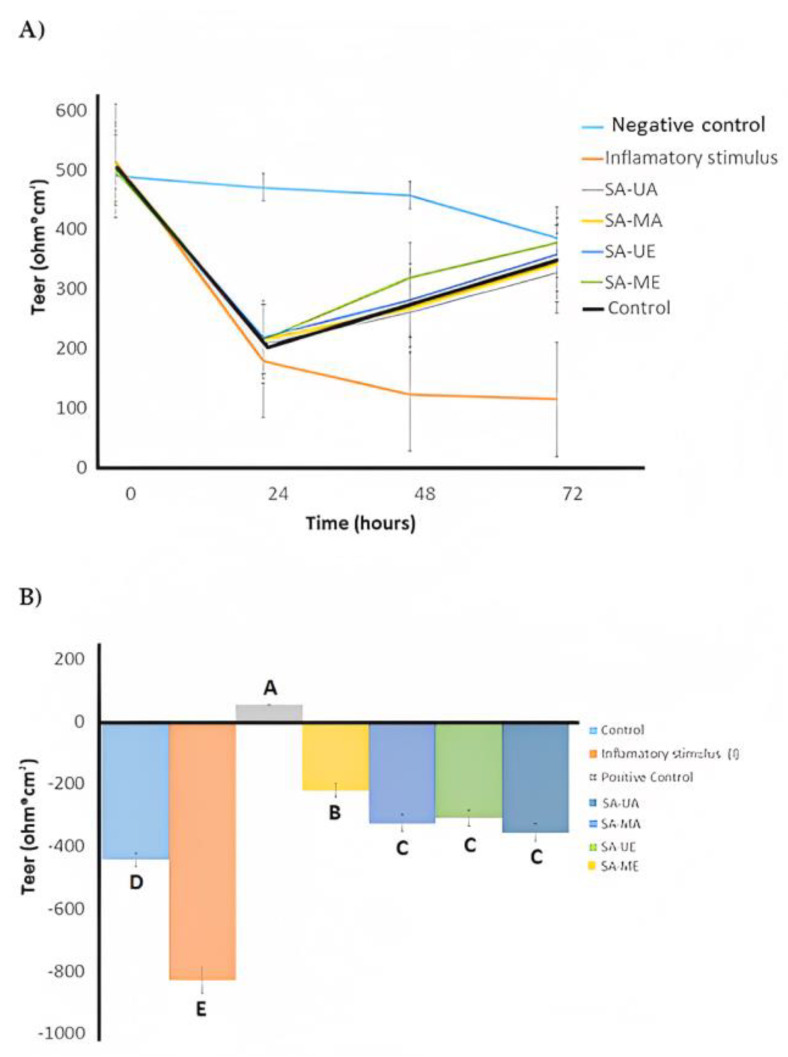
(**A**) Caco-2 cells were treated with negative control (growth media, C), inflammatory stimulus (I, please see text for details) or treated with inflammatory stimulus I and subsequently with control, SA-ME, SA-MA, SA-UE, or SA-UA willow bark extracts after 0, 24, 48 and 72 h. Values are expressed as mean ± SD, *n* = 6, *ρ* < 0.0001 by treatment, time, and their interaction by two-way ANOVA. (**B**) Caco-2 cells treated with control, SA-ME, SA-MA, SA-UE, or SA-UA willow bark extracts for 48 h. Values are expressed as mean ± SD, *n* = 3 for IG, *n* = 15–22 for others. Means with different letters were significantly different as determined by one-way ANOVA followed by Dunnett’s test (*ρ* < 0.05). No significant difference was observed between treatments SA-ME, SA-MA, SA-UE, or SA-UA and the control.

**Figure 4 plants-12-02533-f004:**
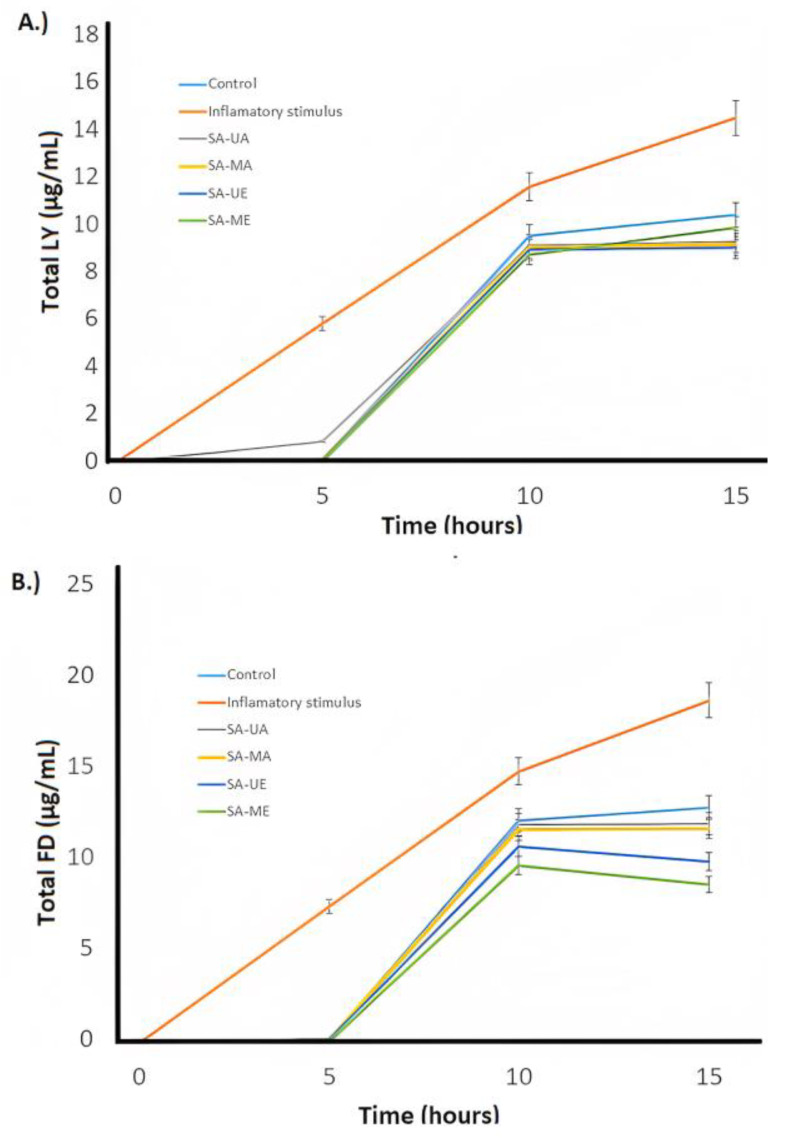
Flux of (**A**) lucifer yellow (LY) and (**B**) fluorescein isothiocyanate-dextran (FD), in differentiated Caco-2 cells exposed to inflammatory stimulus (I) or treated with inflammatory stimulus I and subsequently with control, SA-ME, SA-MA, SA-UE, or SA-UA willow bark extracts for 0, 5, 10 and 15 h. Values are expressed as mean ± SD, *n* = 2–3, where two-way ANOVA was ρ < 0.0001 for time, ρ < 0.1000 for treatment, and <0.0001 for their interaction for FD, and ρ < 0.0001 for time, ρ < 0.1500 for treatment, and <0.0001 for their interaction for LY. Just SA-ME was significantly different compared to other groups regarding FD flux (lower values).

**Figure 5 plants-12-02533-f005:**
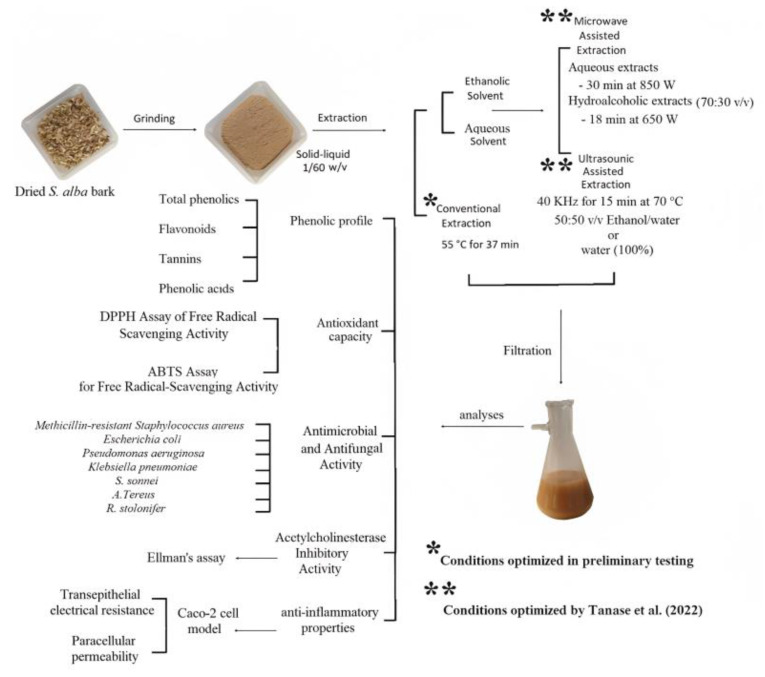
Diagram of experimental design and methods applied with willow (*Salix alba*) bark extracts. * Conditions optimized in preliminary testing. ** Conditions optimized by Tanase et al. (2022) [31].

**Table 1 plants-12-02533-t001:** Total polyphenols content (TPC) and antioxidant activity measured by DPPH• and ABTS•+ methods of willow (Salix alba) bark extracts.

Treatment	TPC (mg/g DW)	DPPH^•^ (μg/mL)	ABTS•+ (μg/mL)
SA-C	136.6 ± 0.8 ^a^	22 ± 1 ^a^	28 ± 2 ^d^
SA-UA	147.6 ± 0.2 ^b^	20 ± 1 ^a^	25 ± 1 ^c^
SA-MA	147.8 ± 0.2 ^b^	17 ± 1 ^b^	22 ± 2 ^b^
SA-UE	158.1 ± 0.9 ^c^	15 ± 1 ^c^	20 ± 1 ^bc^
SA-ME	169.8 ± 0.9 ^e^	12 ± 1 ^d^	16 ± 2 ^a^

Legend: SA-C: willow bark extract obtained via conventional solid–liquid extraction; SA-UA: willow bark extract obtained via ultrasound-assisted extraction with water; SA-MA: willow extract obtained via microwave-assisted extraction with water; SA-UE: willow bark extract obtained via ultrasound-assisted extraction with ethanol; and SA-ME: willow bark extract obtained via microwave-assisted extraction with ethanol. The values are expressed as IC_50_ the concentration of sample (μg/mL) showing 50% of maximal radical scavenging activity for DPPH^•^ and ABTS•+ assays. Values are expressed as mean ± SD (*n* = 3). Different superscripts in the same column are significantly different (*ρ* < 0.05).

**Table 2 plants-12-02533-t002:** Antibacterial and antifungal activity of willow (*Salix alba)* bark extracts.

Inhibition Zone, mm	SA-C	SA-UA	SA-MA	SA-UE	SA-ME
Bacteria					
*S. aureus **	11 ± 1 ^b^	15 ± 2 ^a^	15 ± 1 ^a^	13 ± 1 ^a^	11 ± 1 ^b^
*P. aeruginosa*	8 ± 1 ^b^	10 ± 2 ^a^	7± 1 ^b^	9 ± 1 ^a^	9 ± 2 ^a^
*E. coli*	ND	ND	ND	ND	ND
*K. pneumoniae*	ND	ND	ND	ND	ND
*S. sonnei*	ND	ND	ND	ND	ND
Fungi					
*A. terreus*	ND	ND	ND	ND	ND
*R. stolonifer*	ND	ND	ND	ND	ND

Legend: SA-C: willow bark extract obtained via conventional solid–liquid extraction; SA-UA: willow bark extract obtained via ultrasound-assisted extraction with water; SA-MA: willow extract obtained via microwave-assisted extraction with water; SA-UE: willow bark extract obtained via ultrasound-assisted extraction with ethanol; and SA-ME: willow bark extract obtained via microwave-assisted extraction with ethanol. Values are expressed as mean ± SD (*n* = 3). ^a,b^ Different superscripts in the same row are significantly different (*ρ* < 0.05). ND: Not detected. * methicillin-resistant strain.

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
