# Peer review of "Effect of Microwave and Ultrasound-Assisted Extraction on the Phytochemical and In Vitro Biological Properties of Willow (Salix alba) Bark Aqueous and Ethanolic Extracts"

_plants, 2023, doi:10.3390/plants12132533_

Round 1

Reviewer 1 Report

I hardly see any significant relevant information in the manuscript. A similar work is "Chemical profiling and biological activities of "green" extracts of willow species (Salix L., Salicaceae): Experimental and chemometric approaches" published in Sustainable Chemistry and Pharmacy 32 (2023) 100981.

The article has serious flaws, and the research was not conducted correctly. First of all, the results of the extractions cannot be compared because they do not start from the same concentration of WBP (eg. 1.8% w/v for conventional extraction, 2.5% w/v in UAE, and 2% w/v in MAE).

The authors did not know the described methods: DPPH and Acetylcholinesterase (AChE) inhibitory activity, respectively. In the DPPH method, Trolox is a standard compound, NOT a control. The same observation for AChE inhibition: galantamine is a standard compound, not a positive control.  As a result, the calculation formulas of the inhibitions are wrong.

 The figures are not professionally prepared. 

Author Response

Please see the word document with authors responses.

Reviewer 2 Report

Please refer to the observations/suggestions available as notes in the enclosed file.

Please refer to the observations/suggestions available as notes in the enclosed file.

Author Response

Please see word document with authors responses.

Reviewer 3 Report

The well-presented work is a valuable contribution to the valorization of plant sources for medicinal purposes and thus to the contribution of research in terms of environmental sustainability in line with the goals of Agenda 2030.

There are considerable works in the literature that recall the health value of plant sources for medicinal purposes for example:

Limongelli, F.; Crupi, P.; Clodoveo, M.L.; Corbo, F.; Muraglia, M. Overview of the Polyphenols in Salicornia: From Recovery to Health-Promoting Effect. Molecules 202227, 7954. https://doi.org/10.3390/molecules27227954

Therefore, it is strongly suggested to implement the cited work in the introduction section and also because it is much more recent than those reported.

1.     For the future, it would be interesting to initiate a study of the enriched extracts on cell lines so as to explore the biological activity of the extracts and possibly the synergy of the polyphenols they contain. In this regard, a cytotoxicity study is also suggested.

Author Response

(The authors gave the same response as above.)

Reviewer 4 Report

Dear authors,

Thank you for your valuable manuscript.

The original manuscript it is a comparative study regarding aqueous and ethanolic willow bark extracts obtained by ultrasonic-assisted extraction (UAE) and microwave-assisted extraction (MAE) and analyzed for their phytochemical (total phenolics, phenolic acids, flavonoids, and tannins) content and in vitro biological properties (antibacterial and antifungal activity, acetylcholinesterase AChE inhibitory activity and anti-inflammatory effects).

The experiments presented in manuscript are well conducted. 

I have some suggestions, as follows:

- Line 83: for the obtention of phytochemical-rich extracts, I suggest for phytochemical-rich extracts obtaining” (I marked with yellow)

- Line 102: „Sodium molybdate Fisher Chemical…., I suggest „Sodium molybdate (Fisher Chemical…. (I marked with yellow)

- Line 106: tris/hcl, I suggest Tris/HCl” (I marked with yellow)

- Line 182: 37°C, I suggest 37 °C” (I marked with yellow)

- Line 219: UAe, I suggest UAE” (I marked with yellow)

- Lines 223, 224: (ATCC 25931)), “(ATCC 13429))”, I suggest (ATCC 25931), “(ATCC 13429)” (I marked with yellow)

- Line 244: “405 nm”, I suggest 405 nm wavelength” (I marked with yellow)

- Lines 267, 272: “CO2”, I suggest CO2 (I marked with yellow)

- Line 387 in Table 1:Total polyphenol content”, I suggest Total polyphenols content” (I marked with yellow)

- Lines 409, 413, 414, 490: “μg/ml”, I suggest μg/mL” (I marked with yellow)

- Lines 413, 414: “(μg/ml))”, I suggest (μg/mL)” (I marked with yellow)

- Line 409: gm”, I suggest mg” (I marked with yellow)

- Line 410: 2.19mg”, I suggest “2.19 mg” (I marked with yellow)

- Line 424 and Line 445 in Table 2: “A. Tereus” , I suggest “A. tereus” (I marked with yellow)

- Line 425: “gram-positive” and “gram-negative”, I suggest “Gram-positive” and “Gram-negative” (I marked with yellow)

- Line 445 in Table 2: “Staphylococcus aureus”, for the uniformity of the bacterial strains names, I suggestS. aureus”, as the same it is in the text (I marked with yellow)

- Lines 493, 518: “Caco2”, I suggest “Caco-2” (I marked with yellow)

- Line 495: “cm2”, I suggest “cm2” (I marked with yellow)

- Line 513: “In our study, our hypothesis”, I suggest not 1st person-plural, but 3rd person-neutral, In this study, the proposed hypothesis” (I marked with yellow)

- At References, it is necessary to use Italic Font for the latin names of vegetal species (I marked with yellow)

- At References, I suggest the uniform drafting of citations, in accordance with MDPI requirements.

Thank you!

Minor editing of English language required.

Author Response

Please see word file with authors responses.

Round 2

Reviewer 1 Report

The author replied that he/she has revised the manuscript, but such changes are not significant at all for this paper to be published in Q2 journal.

The author answers that the manuscript has novelty, but he didn’t highlight any element of novelty. These comments are already given in the first review and yet no action was taken by the author.

I understood that you reproduced the MAE and UAE methods described by Tanase et al. (2022). This does not mean that you have compared the efficiency of the extraction of total polyphenols, tannins, and flavonoids by the two methods.

I noticed an improvement in DPPH assay, and also in the enzyme inhibition method in the manuscript.

Author Response

Reviewer 1

The author replied that he/she has revised the manuscript, but such changes are not significant at all for this paper to be published in Q2 journal.

The author answers that the manuscript has novelty, but he didn’t highlight any element of novelty. These comments are already given in the first review and yet no action was taken by the author.

Thank you for this comment, changes have been made in the text in order to highlight the novelty of the present paper (lines 46-50, lines 78-81). Novelty of our study has also been further discussed below.

Salix alba has gained recent interest in the scientific community for its bioactive potential, aspect reflected in the high number of publications surrounding this matrix. Specially, bioactive compound-rich extracts obtained from plant residues such as bark has proved to be an interesting way to promote circular economy and revalorization of the highly produced by-products. However, it is of dire importance to take into consideration the nature of not only the extraction method, which influences the efficiency and quality of the resulting extract, but also the used solvents, as selection should be also focused on their later human consumption for extracts to be of commercial interest. In this sense, most studies do not use green extracts, many of which include commercial extracts and non-GRAS (Generally Recognized As Safe) solvents. Additionally, research is scarce on the use of optimized Salix alba bark extracts obtained through green extraction methodologies (more sustainable and efficient methods to obtain high-quality plant extracts), from which MAE and UAE pose as some of the most promising, and using GRAS solvents, which would allow for the obtention of environmentally friendly extract which additionally are apt for later human consumption. This is a preeminent requirement when considering their potential nutraceutical use. Additionally, the scarce previous literature focused on this topic has not profoundly evaluated the bioactivity and therefore potential of these obtained extracts, and thus, this paper fills a knowledge area which the authors consider necessary to promote research in this direction, for their bioactive purposes.

The most recent-published articles concerning Salix alba are listed below including the main differences with the methodology and results reported in this research: 1) Panaite, T. D., Saracila, M., Papuc, C. P., Predescu, C. N., & Soica, C. (2020). Influence of dietary supplementation of Salix alba bark on performance, oxidative stress parameters in liver and gut microflora of broilers. Animals, 10(6), 958. In this research, conventional extraction was applied using chloroform (not GRAS); 2) Gligorić, E., Igić, R., Čonić, B. S., Kladar, N., Teofilović, B., & Grujić, N. (2023). Chemical profiling and biological activities of “green” extracts of willow species (Salix L., Salicaceae): Experimental and chemometric approaches. Sustainable Chemistry and Pharmacy, 32, 100981. This publication only includes one green extraction method, does not thoroughly evaluate its bioactivity; 3) Bajraktari, D., Bauer, B., & Zeneli, L. (2022). Antioxidant capacity of Salix alba (Fam. Salicaceae) and influence of heavy metal accumulation. Horticulturae, 8(7), 642. Does not apply green extraction methods to obtain the extract; 4) Nica, I. C., Mernea, M., Stoian, G., & Dinischiotu, A. (2020, November). Natural aspirin-like compounds from white willow (Salix alba) bark extract prevent structural changes of human hemoglobin during in vitro non-enzymatic glycation and fructation, preserving its peroxidase and esterase activity. In Medical Sciences Forum (Vol. 2, No. 1, p. 23). MDPI. Does not apply green extraction methods to obtain the extract. 5) Gligorić, E. I., Igić, R., Suvajdžić, L. Đ., Teofilović, B. D., & Grujić-Letić, N. N. (2020). Salix eleagnos Scop.–a novel source of antioxidant and anti-inflammatory compounds: Biochemical screening and in silico approaches. South African Journal of Botany, 128, 339-348. Does not apply green extraction methods to obtain the extract.

In addition, several works have evaluated the biological effects of commercial extracts (which do not allow for the obtained information to be translated into the effect that a more efficient extraction method could have on the bioactivity of the obtained extracts, as it neither includes information for most studies on their phenolic content): 6) Saracila, M., Panaite, T. D., Predescu, N. C., Untea, A. E., & Vlaicu, P. A. (2023). Effect of Dietary Salicin Standardized Extract from Salix alba Bark on Oxidative Stress Biomarkers and Intestinal Microflora of Broiler Chickens Exposed to Heat Stress. Agriculture, 13(3), 698; 7) Saracila, M., Panaite, T. D., Soica, C., Tabuc, C., Olteanu, M., Predescu, C., ... & Criste, R. D. (2019). Use of a hydroalcoholic extract of Salix alba L. bark powder in diets of broilers exposed to high heat stress. South African Journal of Animal Science, 49(5), 944-956; 8) Panaite, T. D., Saracila, M., Papuc, C. P., Predescu, C. N., & Soica, C. (2020). Influence of dietary supplementation of Salix alba bark on performance, oxidative stress parameters in liver and gut microflora of broilers. Animals, 10(6), 958; and 9) Guerrini, A., Dalmonte, T., Lupini, C., Andreani, G., Salaroli, R., Quaglia, G., ... & Isani, G. (2022). Influence of dietary supplementation with Boswellia serrata and Salix alba on performance and blood biochemistry in free-range leghorn laying hens. Veterinary Sciences, 9(4), 182.

In the submitted paper, potential of Green and GRAS extracts obtained through innovative advanced technologies on total phenolic content, total flavonoid content, tannin content, antioxidant activity (measured through DPPH and ABTS+ assays), antimicrobial and antifungal activities, acetylcholinesterase inhibitory activity and anti-inflammatory activity on a Caco-2 culture were evaluated. Assessment of the mentioned bioactive properties on Salix alba bark MAE and UAE extracts puts forward new information on the presented field that will allow for the development of future further and more specific research.

I understood that you reproduced the MAE and UAE methods described by Tanase et al. (2022). This does not mean that you have compared the efficiency of the extraction of total polyphenols, tannins, and flavonoids by the two methods.

Thank you very much for your comment. MAE and UAE conditions described by Tanase et al. (2022) were reproduced as preliminary conditions for obtaining bioactive enriched extracts in order to achieve an exploratory evaluation of the bioactive potential. In this way, the combination of these extraction techniques and the evaluation of their biological properties could provide new information since no previous research has focused on the here studied area. In this sense, and due to the initial nature of the study, phenolic, flavonoid and tannin content were analysed with the intention of correlating the observed bioactivity for each assay with content of these compounds. Due to the positive results of our study, additional analysis including an optimization of these parameters for the selected plant matrix will be considered for further studies as a way to observe the potential improvement further efficacy of extraction could have on bioactive properties evaluated in this paper.

I noticed an improvement in DPPH assay, and also in the enzyme inhibition method in the manuscript.

Thank you for your comment.

Reviewer 2 Report

There are still needed some modifications before the manuscript will be suitable for publication.

Please refer to the notes in the enclosed file.

N/A

Author Response

Reviewer 2

There are still needed some modifications before the manuscript will be suitable for publication.

Please refer to the notes in the enclosed file.

Thank you for your revision. Corrections were made as suggested throughout the text.
